# SIG-CHAT: SPATIAL INTENT-GUIDED CONVERSATIONAL GESTURE GENERATION INVOLVING HOW, WHEN AND WHERE

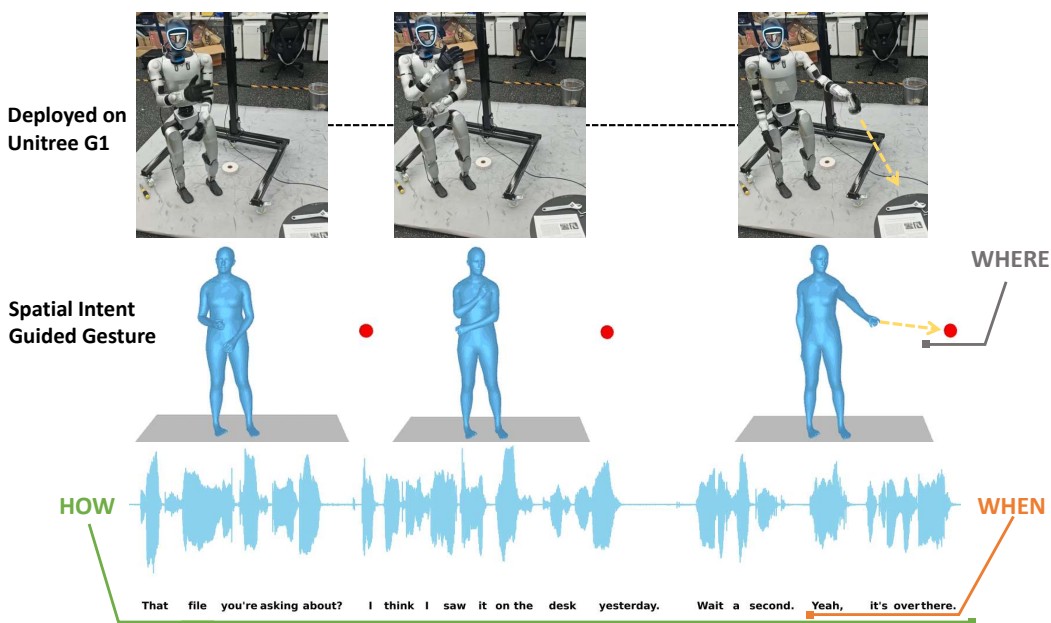

Figure 1: Driven by audio and spatial intent information, interactive conversational gestures can be generated. Audio determines HOW non-interactive gestures are produced and also WHEN the intent-driven interactions are conducted, such as when referring to an object with phrases like "It's over there." Meanwhile, spatial intent guides WHERE to interact, including the torso and head orienting toward the referent, while the hand pointing toward the target. When deployed on a humanoid robot, this framework allows for conversational gestures capable of interacting with the environment.

## ABSTRACT

The accompanying actions and gestures in dialogue are often closely linked to interactions with the environment, such as looking toward the interlocutor or using gestures to point to the described target at appropriate moments. Speech and semantics guide the production of gestures by determining their timing (WHEN) and style (HOW), while the spatial locations of interactive objects dictate their directional execution (WHERE). Existing approaches either rely solely on descriptive language to generate motions or utilize audio to produce non-interactive gestures, thereby lacking the characterization of interactive timing and spatial intent. This significantly limits the applicability of conversational gesture generation, whether in robotics or in the fields of game and animation production. To address this gap, we present a full-stack solution. We first established a unique data collection method to simultaneously capture high-precision human motion and spatial intent. We then developed a generation model driven by audio, language, and spatial data, alongside dedicated metrics for evaluating interaction timing and spatial accuracy. Finally, we deployed the solution on a humanoid robot, enabling rich, context-aware physical interactions. Our data, models, and deployment solutions will be fully released.

# 1 INTRODUCTION

Human dialogue is more than just words. We naturally use gestures and body language to express intentions and meanings that speech alone cannot convey. These nonverbal signals play a critical role in communication. For instance, during multi-party conversations, individuals frequently orient their heads and torsos toward different speakers while keeping their lower body stationary. Similarly, when describing objects or events situated in space, people often gesture toward relevant locations to ground their references. Therefore, beyond HOW they are physically produced, human actions during conversation inherently carry important WHEN and WHERE information, clarifying both timing and spatial intent.

The growing demand for conversational action generation spans applications in gaming, animation, virtual social platforms, and humanoid robotics. While prior work has produced kinematically natural gestures synchronized with speech rhythm Ao et al. (2022) or enriched by semantic and speaker context Yang et al. (2023a); Zhi et al. (2023), these methods typically rely solely on speech input. Without contextual spatial guidance, they often fail to engage in spatially appropriate interactions, focusing on how to gesture while neglecting when and where to interact. Although several methods generate interactive motions in non-conversational settings Liang et al. (2024); Xu et al. (2024); Yin et al. (2023b), they generally respond to another person's actions, lack spatial generalization, and omit audio, making them unsuitable for conversational scenarios. Others, such as WANDR Diomataris et al. (2024), focus on intent-guided locomotion rather than communication. Therefore, generating intent-aware gestures that interact accurately within specific spatial and temporal contexts remains a critical open challenge.

To bridge this gap, we extend conversational motion generation by integrating environmental spatial signals. Specifically, we incorporate the most common visual and deictic pointing intents, using the target object's 3D position relative to the root joint as a spatial cue. This cue, combined with speech and other inputs, enables the generation of spatially grounded and contextually appropriate gestures. Here, **visual intent** refers to gaze direction toward the interlocutor or referenced objects, while **deictic pointing intent** denotes hand gestures that explicitly indicate objects or directions mentioned in speech. For example, given the audio "*That file you're asking about?... it's over there*" and the spatial location of a file, the model can generate a thinking gesture followed by pointing toward the file's position when referenced, as shown in Figure 1.

Advancing in this direction, we present **SIG-Chat**, a large-scale motion-captured dataset for dialogue gestures, augmented with intent-aware spatial data. Using Xsens for motion capture and SteamVR with HTC Vive for target localization, all signals are synchronized into a unified coordinate system via MVN software. The dataset contains 7,723 clips ($\approx$ 9 hours) from 6 speakers, annotated with three spatially interactive intents: *visual focus*, *left-hand pointing*, and *right-hand pointing*. These are defined as one-hot categories due to differing evaluation metrics for gaze and pointing. Each clip includes synchronized audio, transcript, intent category, and time-aligned 3D target position or trajectory. Spatial intents were designed with diverse locations per utterance, incorporating both static positions and dynamic trajectories. We also develop a baseline model based on the Diffusion Transformer (DiT) to synthesize spatially interactive gestures. It integrates multimodal fusion modules to align gesture kinematics with speech rhythm, semantic content, and spatial intent targets.

We introduce a comprehensive benchmark for this task. Audio-motion alignment is measured by Beat Consistency Score (BC), while temporal and spatial accuracy of interactions are evaluated using our several new metrics. One of them is the Intent Alignment Ratio (IAR@$k$) metric, which calculates the proportion of poses where the interactive intent is successfully executed within an orientation deviation of $k°$. We also build a full pipeline for humanoid robot deployment, integrating visual perception, and motion retargeting modules.

Overall, our contributions can be summarized as follows:

- We construct SIG-Chat, a multimodal speech-gesture dataset integrating intent annotations and 3D spatial metadata of intent targets, providing a rich foundation for intent-aware gesture synthesis.

- We propose a baseline model that accepts as inputs audio, transcript, initial posture description, and trajectory of intent target to synthesize spatially interactive co-speech gestures through multimodal fusion modules.

- We establish a benchmark for intent-aware conversational gesture synthesis, introducing three metrics that explicitly evaluate the temporal and spatial accuracy of interactions driven by multimodal inputs.

## 2 RELATED WORK

### 2.1 CONVERSATIONAL GESTURE GENERATION

The field has evolved from rule-based systems to multimodal neural networks. Early methods used keyword-triggered rules Habibie et al. (2022) or statistical models on datasets like Trinity Ferstl & McDonnell (2018), yielding basic yet repetitive synchronization. Rhythmic coordination improved via prosodic hierarchies (Rhythmic Gesticulator Ao et al. (2022)) and hybrid models (AQ-GT Voß & Kopp (2023)) trained on rhythm-annotated data (BEAT Liu et al. (2022)). Diffusion models with kinematic constraints Alexanderson et al. (2023) further reduced latency. Recent work incorporates semantics and style: emotion-aware generation uses affective labels (EMoG Yin et al. (2023a)) or CLIP guidance (SaGA++ Kucherenko et al. (2021)), while speaker-specific datasets (PATS Ahuja et al. (2020)) enable personalized encoding (MPE4G Kim et al. (2023)). Style transfer leverages cultural tags (Gesture-Speech 3D Habibie et al. (2021)) or reference clips (ZeroEGGS Ghorbani et al. (2023)) for zero-shot adaptation. Despite progress in co-speech gesture generation, a critical gap remains in incorporating spatial awareness and intent to drive conversational interactive gestures.

### 2.2 CONTEXT-AWARE MOTION GENERATION

Recent context-aware motion generation integrates environmental, social, and task constraints to synthesize adaptive behaviors. Methods include multi-agent interaction models Zuo et al. (2023); Lee et al. (2024), geometry-aware navigation Huang et al. (2023), and goal-directed systems like WANDR Diomataris et al. (2024). Multimodal transformers Lou et al. (2024); Wang et al. (2021) align motion with scenes, while social perception approaches Deichler et al. (2022); Yang et al. (2023b) encode cultural styles. Yet, spatial perception remains underdeveloped for unseen interaction scenarios. WANDR Diomataris et al. (2024), for instance, models wrist-level motion and body locomotion without speech, lacking timing cues for interaction. We propose modeling not only how gestures are performed but also when and where to interact, guided by audio and spatial intent.

### 2.3 CONDITIONAL DIFFUSION MODEL

Denoising diffusion probabilistic models (DDPMs) Ho et al. (2020); Song et al. (2021) have advanced conditional synthesis by progressively integrating diverse guidance signals. Early frameworks like Stable Diffusion Rombach et al. (2022) utilized CLIP embeddings for text-to-image generation, later extended to video Ho et al. (2022) and 3D domains Poole et al. (2023). Architectural innovations such as ControlNet Zhang & Agrawala (2023) introduced fine-grained spatial control via pose, edge, and depth adapters, while temporal conditioning enabled camera motion synthesis in videos Wang et al. (2023). Recent efforts unify contextual inputs (e.g., maps, planned actions) for scenario-aware prediction in specialized domains Wang et al. (2024), reflecting a paradigm shift toward multi-modal conditioning across spatial, temporal, and semantic dimensions.

## 3 DATASET

### 3.1 OVERVIEW AND DATA DESIGN

This section presents the SIG-Chat dataset, which extends beyond internal speaker traits (e.g., emotion, style) by incorporating spatial intent annotations. Each sample includes a 3D gesture sequence, synchronized audio and transcript, intent category, initial pose description, and the 3D position or trajectory of the intent target. The dataset is partitioned into two tracks based on whether utterances contain explicit linguistic phrases for spatial interaction, like 'over there' or 'look at this'. Specifically, Track-I consists of rich conversational gestures where interactions are more spontaneous and contextually driven, without explicit interactive phrases. Therefore, these gestures lack an explicit temporal correlation between the audio and the interaction. In contrast, Track-II comprises utterances

Table 1: Comparison with other human motion datasets. Noticeable, spatial intents in our dataset include both static locations and dynamic trajectories.

| Dataset | Clips | Modalities | Motion Representation | Motion Acquisition |
|---|---|---|---|---|
| AMASS Mahmood et al. (2019) | 11265 | 3D Motion | SMPL | Mocap |
| BAREL Punnakkal et al. (2021) | 13220 | Text, 3D Motion | SMPL | Mocap |
| HumanML3D Guo et al. (2022) | 14616 | Text, 3D Motion | SMPL | Mocap |
| MotionX Lin et al. (2023) | 81084 | Text, 3D Motion | SMPL | Pseudo label |
| BEAT Liu et al. (2022) | 2508 | Speech Audio, 3D Motion | BVH | Mocap |
| BEAT2 Liu et al. (2024) | 1762 | Speech Audio, 3D Motion | SMPLX | Mocap |
| AIST++ Li et al. (2021) | 1408 | Music Audio, 3D Motion | SMPL | Mocap |
| **SIG-Chat (Ours)** | 7723 | Text, Speech Audio, **Spatial Intent**, 3D Motion | SMPLH | Mocap |

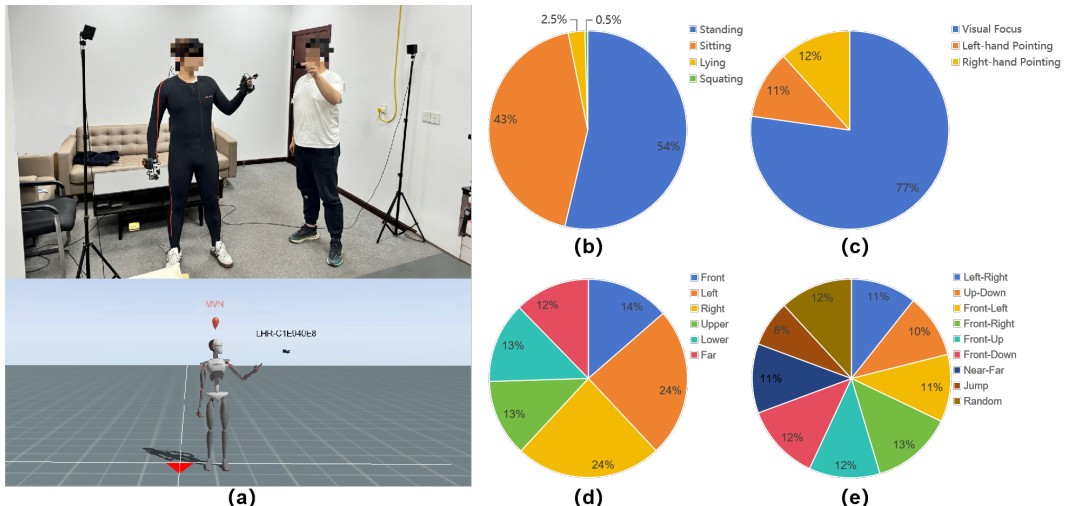

Figure 2: **Capture System and Distribution of SIG-Chat.** (a) Synchronized Gesture-Inten Recording via MVN motion capture system with HTC Vive. (b) SIG-Chat incorporates 4 initial states. (c) Speakers' gestures are classified into 3 intent categories. (d) Type distribution of static target positions. (e) Type distribution of dynamic target trajectories.

containing explicit linguistic phrases. The explicit nature of these utterances creates a strong temporal correlation between the audio and the gesture interaction, allowed for the annotation of precise interaction time spans. With these time span annotations, the temporal accuracy of the generated gestures can be quantitatively evaluated.

**Data Statistics** The dataset contains 7,123 gesture sequences (60M frames, 8.4 hours), covering 15 spatial patterns across three intent categories(gaze, left&right hand pointing). Utterances are in Chinese and English (4:6 ratio), with durations of 1–60 seconds. The larger Track-I contains 6,009 sequences, providing a rich corpus for learning robust, context-driven gestures, while the more focused Track-II contains 1,114 sequences, enabling the model to master temporally precise interactions. As shown in Figure 2.c, gestures are categorized into three intents: visual focus (77%, 6.6h), left-hand pointing (11%, 0.9h), and right-hand pointing (12%, 0.9h). Intent targets exhibit both static positions and dynamic trajectories. Static positions (Figure 2.d) cover regions including Front (14%), Left (24%), Far (12%) and etc. Dynamic trajectories (Figure 2.e) include transitions of intent target such as Left2Right (11%), Up2Down (10%), Near2Far (11%) and etc. The dataset also incorporates four initial body states (sitting, standing, squatting, lying) as shown in Figure 2.b.

## 3.2 DATA COLLECTION

As illustrated in Figure 2(a) (top), data is captured in a 2m × 2m area using (1) Xsens inertial suit for body motion (23 joints), (2) Manus gloves for hand articulation (30 joints), and (3) SteamVR Stations paired with HTC Vive for intent target localization. MVN software synchronizes all streams

into a unified coordinate space (see Figure 2 (a) bottom). Motion sequences are retargeted from Xsens (23+30 joints) to SMPL-H (22+30 joints) via Unreal Engine 5, ensuring kinematic validity and compatibility with industry standards. Temporal alignment is achieved by down-sampling all motion sequences to 20 FPS. When capturing, one person holds the HTC Vive location sensor to generate static or dynamic spatial intent data, the actor speaks and performs gestures according to the spatial intent and the content said.

### 3.3 WHY USE SPATIAL RATHER THAN VISUAL SIGNALS?

A natural question is why we use spatial locations rather than end-to-end learning from direct visual signals. The reason lies in robustness and generalizability: an end-to-end approach would tightly couple the model to the camera's extrinsic parameters. Changing the camera position or robot platform would break the system. By using decoupled spatial locations, the model remains functional through simple recalibration of the camera-to-robot transformation—essential for reliable, generalizable deployment.

## 4 METRICS

We propose an evaluation scheme for intent-aware gestures by modeling the geometric relationship between skeletal motion and intent targets. Using key SMPL-H joints, we derive an intent execution direction (e.g., head orientation) and an expected target direction (e.g., eye-to-target vector).

The Intent Angular Deviation (IAD) metric calculates the angular deviation $\theta$ per frame between these directions: for visual intent, IAD measures the angular deviation between facial orientation and the gaze vector to the target; for pointing intent, it uses the pointing limb direction and the target vector (see Appendix B for more details). Based on IAD, which is denoted as $\theta$, we introduce two main metrics:

Table 2: The IAD and IAR metrics for ground-truth (human) interactive behaviors. The statistical data is selected from gestures within interaction timestamps in SIG-Chat Track-II subset.

| Visual Intent Gesture | | | | Pointing Intent Gesture | | | |
|---|---|---|---|---|---|---|---|
| IAD(mean $\pm$ std) | IAR@25° | IAR@30° | IAR@35° | IAD(mean $\pm$ std) | IAR@10° | IAR@15° | IAR@20° |
| 16.658 $\pm$ 9.192 | 0.630 | 0.808 | 0.943 | 9.623 $\pm$ 6.316 | 0.646 | 0.896 | 0.953 |

### 4.1 INTENT ALIGNMENT RATIO (IAR@$k$)

IAR calculates the average proportion of frames where the interactive intent is successfully executed within Intent Angular Deviation thresholds $k$:

$$\text{IAR@}k = \frac{1}{N}\frac{1}{T}\sum_{i=1}^{N}\sum_{t=1}^{T}\mathbb{I}(\theta_{i,t} \le k) \tag{1}$$

where $N, T$ represent the number of sequences and frames, $\theta_{i,t}$ denotes the IAD for the t-th frame of the i-th motion, $\mathbb{I}$ is the indicator function.

In practice, human intent in spatial referencing is not always precisely aligned with the target. Consequently, the IAD for ground-truth human motions (in Table 2) is inherently non-zero. To establish spatial intent evaluation thresholds that account for this natural variance, we first calculate the IAD distribution on the ground-truth data. As shown in Table 2, the mean IAD for visual and pointing intents are $16.658 \pm 9.192$ and $9.623 \pm 6.316$, respectively. The threshold $k$ for the IAR metric is then determined based on the $mean + std$ of the IAD values, reflecting realistic tolerance levels. Specifically, the angular thresholds are set to $15°$ for pointing and $30°$ for gaze. This choice is further validated by the ground-truth IAR scores in Table 2.

Figure 3: Architecture of the denoising network. The model is a multi-layer diffusion-transformer with two fusion modules (speech fusion and intent fusion) that integrates audio, transcript, initial posture description, intent category, and target trajectory as multimodal contitions to estimate the clean gesture $\hat{x}_0$ from noisy gesture $x_t$ at diffusion timestep $t$. Multimodal inputs are processed and encoded by specific encoders. FC refers to the Fully Connected (FC) layer.

## 4.2 THRESHOLDED INTERSECTION-OVER-UNION(IOU@$k$)

The IOU@$k$ is designed to evaluate the temporal accuracy of interactions in Track-II. Data in this track is annotated with the valid time span of executing intent. It is defined as follow:

$$\text{IoU@}k = \frac{|T@k \cap T_{\text{gt}}|}{|T@k \cup T_{\text{gt}}|} \tag{2}$$

where $T@k = \sum_{t=1}^{T} \mathbb{I}(\theta_t \leq k)$. During the evaluation of Track-II, the IAR@$k$ is only calculated in the time span of $T_{\text{gt}}$.

## 5 INTENT-AWARE GESTURES SYNTHESIS BASELINE

In this part, we propose an intent-aware gesture synthesis baseline for generating spatially interactive gestures with intent guidance. As depicted in Figure 3, our approach builds on the Diffusion-Transformer (DiT) framework by integrating multimodal fusion modules. These modules are key to dynamically synchronizing the gesture's motion with speech rhythms, linguistic meaning, and the spatial targets of the user's intent. The network learns to denoise a gesture sequence $x_t$ at a given diffusion timestep $t$, guided by a rich set of conditioning information: the speaker's audio and transcribed text, the category of intent, the initial body pose, and the 3D paths of any target objects. The architecture is composed of two main stages: (1) multimodal encoders (Section 5.1) that project the different input types into synchronous feature space, and (2) a gesture synthesis backbone (Section 5.2) featuring adaptive fusion mechanisms to merge the conditioning features with the gesture representation at each step of the diffusion process. Below, we elaborate on our input processing methods and the design of the DiT backbone.

### 5.1 MULTIMODAL ENCODERS

**Gesture Representation.** Each skeleton posture $p_i$ in the gesture sequence $x = \{p_1, ..., p_N\}$ of $N$ frames is represented by root translation and joint rotations. Specifically, $p_i = (r^x, r^y, r^z, j^r)$, where $(r^x, r^z) \in \mathbb{R}^2$ denotes horizontal root translation, $r^y \in \mathbb{R}$ is root height and $j^r \in \mathbb{R}^{6 \times J}$ represents 6D continuous rotations for $J$ joints. The gesture input is subsequently linearly projected into the embedding space, denoted as $\mathbf{A} \in \mathbb{R}^{N \times 256}$.

**Intent-Trajectory Joint Representation.** The intent category is encoded as a one-hot vector, and the target trajectory is represented by a 3D global translation sequence $\mathbb{R}^{N \times 3}$. By concatenating them along the feature dimension, we get a composite input $\mathbb{R}^{N \times (3+n)}$ where $n$ is the number of intent categories. It is linearly projected into latent space, yielding the intent-target feature $\mathbf{I} \in \mathbb{R}^{N \times 64}$ for intent fusion. The choice of a one-hot representation over a latent code is due to the distinct evaluation criteria required for each intent type.

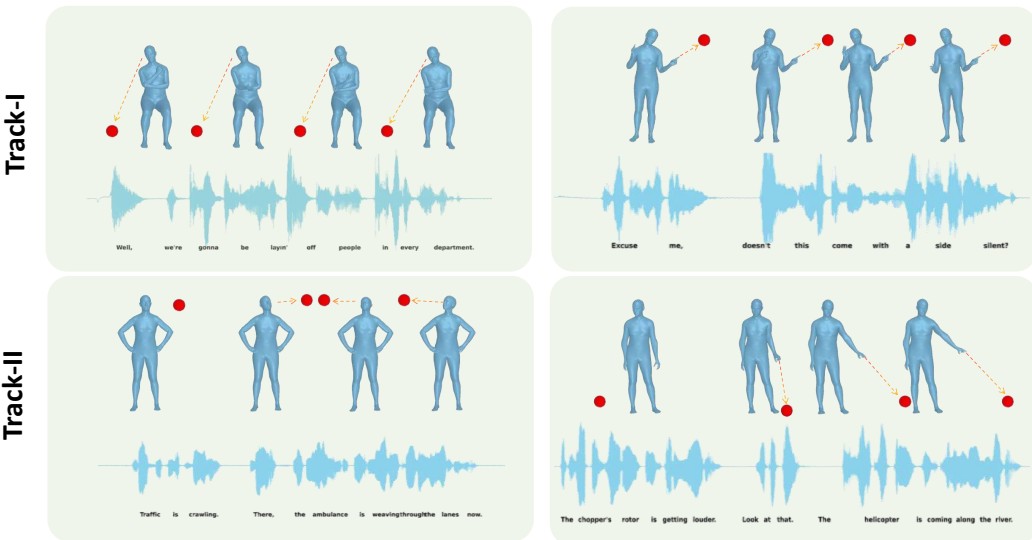

Figure 4: Visualization results of intent-aware gesture generated by SIG-Chat.

**Diffusion Step.** Following Rombach et al. (2022), the diffusion step $t$ is encoded through positional encoding and nonlinear projection to produce timestep embedding.

**Audio.** Audio recordings are first processed by the WavLM Large pre-trained model Chen et al. (2022) to extract contextual rhythm features. The resulting feature sequence is temporally synchronized to the 20-fps gesture timeline via linear interpolation and subsequently projected into an audio embedding space, forming the representation $\mathbf{A} \in \mathbb{R}^{N \times 256}$, which provides rhythm information for Speech-Aware Attention.

**Transcript.** Transcripts are synchronized with audio using Gentle Ochshorn & Hawkins (2017) and mapped to the gesture timeline. Words are embedded with FastText Bojanowski et al. (2017) and linearly projected to the transcript feature $\mathbf{T} \in \mathbb{R}^{N \times 256}$.

**Initial Posture Description.** Descriptions of initial postures are encoded using the text encoder of CLIP ViT-B/32 Radford et al. (2021) to extract sentence-level embeddings $\mathbf{P} \in \mathbb{R}^{512}$.

### 5.2 GESTURE SYNTHESIS BACKBONE

Inspired by the cross-modal fusion strategy in StableMofusion Huang et al. (2024), our denoising network employs a DiT backbone with Adaptive Layer Normalization (AdaLN) and zero initialization. To achieve hierarchical alignment of gesture embedding sequences with multimodal inputs, it incorporates two fusion modules, as illustrated in Figure 3.

The **Speech Fusion Module** fuses speech (audio and transcript) features and initial posture features into gesture representations. It first concatenates temporally synchronized speech features with gesture hidden states, projects them via a linear layer, and prepends an initial posture token to the combined sequence. A multi-head attention mechanism is then applied to capture alignment between speech rhythms/semantics and gesture kinematics.

The **Intent Fusion Module** further refines the output states by incorporating spatial-temporal intent targets. The Gesture states are merged with temporally synchronized intent features through concatenation and linear projection, followed by a multi-head attention mechanism to model kinematic-intent dependencies.

### 5.3 MIXTURE TRAINING STRATEGY

Our dataset contains two subsets, Track-I and Track-II, which are highly imbalanced in size and semantic explicitness. Uniform sampling would bias the model toward the more frequent but less interaction-explicit samples in Track-I, impairing its ability to learn spatially interactive gestures

synchronized with audio cues. To mitigate this, we adopt a mixture batch resampling strategy with an 8:2 sampling ratio from Track-II to Track-I. Each batch is constructed without replacement per epoch, forcing the model to prioritize interaction-rich examples from Track-II while maintaining diversity from Track-I. This up-sampling of Track-II ensures sufficient exposure to explicit interaction semantics while preserving generalization ability through Track-I's varied gestures.

## 5.4 Deployment Solution on Humanoid Robots

The pipeline decomposes 6D rotations into axis–angle form, aligns root orientations, and packages the data following AMASS format. We firstly use Mink retargeting Xie et al. (2025) to map SMPL motions to the Unitree G1 via IK-based optimization, enforcing joint limits and end-effector consistency while filtering for physical plausibility. Then, the retargeted motions are fed into off-the-shelf motion tracking policy. The resulting policies are integrated into the robot's configuration modules to generate stable, executable actions on the Unitree G1. Visual perception is handled by YOLOWorld Cheng et al. (2024).

## 6 Experiments

Table 3: Quantitative results on gestures with visual intent on the SIG-Chat test set.

|  | Method | IAR@30°↑ | IOU@30°↑ | FGD↓ | BC SCORE↑ | Diversity↑ |
|---|---|---|---|---|---|---|
| Track-I | GT | 0.792 | - | - | - | - |
|  | ours (w/o intent) | 0.454 | - | 20.611 | 0.925 | 15.416 |
|  | ours (w/ intent) | 0.796 | - | 16.444 | 0.919 | 15.162 |
| Track-II | GT | 0.808 | - | - | - | - |
|  | ours (w/o intent) | 0.473 | 0.36 | 16.762 | 0.944 | 12.033 |
|  | ours (w/ intent) | 0.710 | 0.545 | 6.716 | 0.944 | 11.209 |

Table 4: Quantitative results on gestures with pointing intent on the SIG-Chat test set.

|  | Method | min IAD↓ | IAR@15°↑ | IOU@15°↑ | FGD↓ | BC SCORE↑ | Diversity↑ |
|---|---|---|---|---|---|---|---|
| Track-I | GT | 8.178 | 0.507 | - | - | - | - |
|  | ours (w/o intent) | 46.857 | 0.06 | - | 70.325 | 0.93 | 15.107 |
|  | ours (w/ intent) | 10.459 | 0.358 | - | 42.526 | 0.924 | 14.14 |
| Track-II | GT | 4.997 | 0.896 | - | - | - | - |
|  | ours (w/o intent) | 47.75 | 0.204 | 0.164 | 15.711 | 0.935 | 12.917 |
|  | ours (w/ intent) | 5.153 | 0.846 | 0.637 | 1.141 | 0.941 | 13.419 |

## 6.1 Evaluation of the Baseline Model

**Metrics.** We evaluate the spatial and temporal accuracy of interactive gestures using our proposed metrics (see Section 4). For pointing gestures, we measure peak spatial precision with the minimum Intent Angular Deviation (**min IAD**) and Intent Accuracy Rate (**IAR@k**), the percentage of correctly aimed frames. Temporal accuracy for both intents is evaluated using Intersection over Union (**IOU@k**), the temporal overlap of correct gesture segments. The angular thresholds are set to 15° for pointing and 30° for gaze, respectively (according to the statistical results presented in Section 4.1). Beyond our proposed metrics, we employ Fréchet Gesture Distance (**FGD**) Yoon et al. (2020) to assess the realism and naturalness of generated gestures by training a transformer auto-encoder to extract gesture features. **Diversity** is measured as the average L1 distance between features of multiple generated gestures. Beat Consistency Score (**BC**) Li et al. (2021) is used to assess speech-motion synchrony.

**Quantitative Results.** As presented in Table 3 and Table 4, our proposed intent-aware model demonstrates a substantial quantitative advantage over the w/o intent baseline, which lacks explicit intent modeling. For visual intent (Table 3), our model achieves a dramatic increase in IAR, from

0.473 to 0.710 in Track-II, closely approaching the ground truth performance. A similar, pronounced trend is observed for pointing gestures (Table 4), where our model significantly lowers min IAD and boosts the IAR, confirming a marked improvement in pointing precision. These results collectively validate the efficacy of our approach in generating high-fidelity, spatially interactive chat gestures.

## 6.2 ABLATION STUDY

To validate our key design choices, we conduct a series of ablation studies, with detailed results provided in the Appendix. We first benchmark our model against state-of-the-art methods on BEATv2 to demonstrate its competitive performance (Appendix C.1). We then confirm the benefits of our mixture batch sampling scheme over uniform sampling (Appendix C.2), the effectiveness of our multimodal fusion architecture (Appendix C.3), and the spatial representation experiments of intent target (Appendix C.4).

## 6.3 VISUALIZATION RESULTS

The visualization results as shown Figure 4 illustrates the ability of our method to generate conversational gestures based on semantic and spatial intent (direction or trajectory), resulting in precisely timed and accurate interactions with objects in the spatial environment. In Track-I, the utterance semantics contain no explicit lexical triggers for interaction; here, environmental engagements are generated naturally through implicit cues. In Track-II, the semantics include clear triggering phrases, which explicitly guide the model to produce spatially intentional interactions at appropriate moments.

## 6.4 USER STUDY

We asked 20 users to compare 50 pairs of samples. It compares our model with an ablated version without our intent controls, and also compares our model with the ground truth motion. Each pair of videos corresponds to the same audio and intent goal. For each pair, we ask two questions: Q1 (Perceptual Accuracy): "Which video shows the speaker looking at or pointing to the target better?" Q2 (Speech-Gesture Fluency): "Which video shows more fluent/natural gesture-speech coordination?" The aggregated preference rates are summarized in the Table 5.

Table 5: User study on the audio to interactive gesture for the SIG-Chat dataset

| Comparison | Question | Our Model | Competitor | Tie/Same |
|---|---|---|---|---|
| Ours vs. w/o Intent | Q1:Perceptual Accuracy | 78.7% | 11.1% | 10.2% |
| Ours vs. w/o Intent | Q2:Speech-Gesture Fluency | 70.8% | 15.7% | 13.5% |
| Ours vs. Ground Truth | Q1:Perceptual Accuracy | 30.0% | 40.8% | 29.2% |
| Ours vs. Ground Truth | Q2:Speech-Gesture Fluency | 37.3% | 38.5% | 24.2% |

The results clearly validate our method vs. w/o Intent, our model overwhelmingly wins in both perceptual accuracy and speech-gesture fluency. This demonstrates the significant and positive impact of our proposed intent-aware module. Compared with GT, our model's performance is competitive. It achieves a near-equal preference rate on speech-gesture against the GT. This confirms our model's ability to generate high-quality, perceptually correct gestures. We will include the full details of this user study in our revision. Thank you again for helping us improve the paper's adequacy of assessment.

## 7 CONCLUSION

We introduce spatially intent-guided gesture generation, a task extending speech synchronization to spatial intent, requiring control over gesture style (HOW), timing (WHEN), and spatial target (WHERE). To support this, we collect a novel dataset co-recording gestures with speech and spatial data, develop a baseline model with benchmark metrics, and demonstrate deployment on a humanoid robot. We hope this work invigorates gesture research and advances more intelligent, anthropomorphic robots.

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

# APPENDIX

## A  IMPLEMENT DETAILS

The SIG-Chat dataset is divided into training, validation, and test sets in a ratio of 0.8: 0.05: 0.15. For training the diffusion model, we use DDPM with $T = 1,000$ diffusion steps and variances $\beta_t$ linearly from 0.0001 to 0.02 in the forward process. And we use AdamW with an initial learning rate of 5e-5 and a 0.01 weight decay to train the denoising model for 60,000 iterations at a batch size of 64. On gradient descent, clip the gradient norm to 1. It takes about 13 hours for training on an RTX A100. All evaluations are conducted with 10 runs, and it takes about 1.5 hours for the evaluation with 32 batch size on an RTX A100.

## B  INTENT ANGULAR DEVIATION

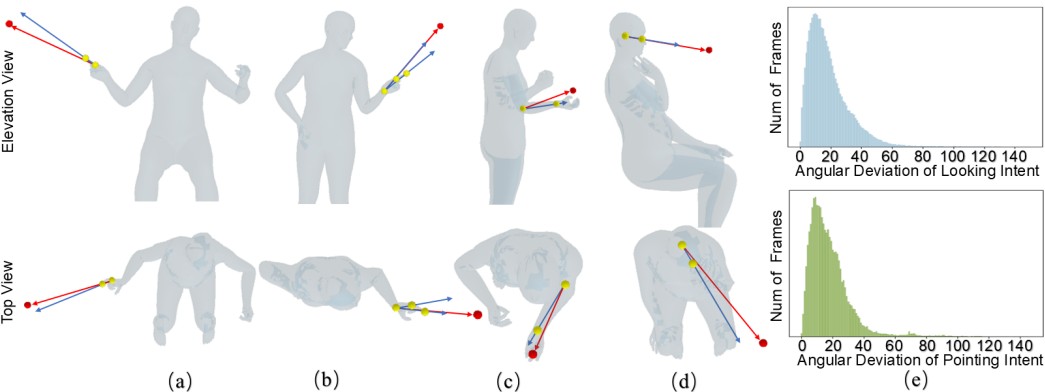

Figure 5: Angular deviation calculation diagrams (a–d) show Index Finger, Hand, Arm, and Gaze Orientation Deviations, with red spheres for intent targets, yellow for key joints, blue arrows for gesture intent directions, and red arrows for expected target direction. Histograms (e) presents statistics of gazing/pointing angular deviations in the SIG-Chat dataset.

IAD calculates the angular deviations $\theta$ between the generated skeleton's key joint direction vectors and the target direction vector for each frame, capturing both gaze intent and pointing intent.

$$\text{IAD} = \frac{1}{N}\frac{1}{T}\sum_{i=1}^{N}\sum_{t=1}^{T}|\theta| \tag{3}$$

where $N$,$T$ represent the number of data and frames, respectively. For pointing intent, the angular deviation $\theta$ is determined by the minimum angle among three specific directional vectors: $\theta = \min(\theta_{\text{finger}}, \theta_{\text{arm}}, \theta_{\text{cone}})$:

1. Index Finger Orientation Deviation (Figure 5 (a)) which measures the angular difference between the vector $\vec{v}_{\text{tip-base}}$ originating from the base of the index finger to its tip and the vector $\vec{v}_{\text{base-target}}$ extending from the base of the index finger to the intended target: $\theta_{\text{finger}} = \arccos\left(\frac{\vec{v}_{\text{tip-base}} \cdot \vec{v}_{\text{base-target}}}{||\vec{v}_{\text{tip-base}}|| ||\vec{v}_{\text{base-target}}||}\right)$.

2. Hand Orientation Deviation (Figure 5 (b)) defined as the angular difference between the wrist-hand vector $\vec{v}_{\text{wrist-tip}}/\vec{v}_{\text{wrist-base}}$ (either from the wrist to the base of the index finger or from the wrist to its tip) and the vector $\vec{v}_{\text{wrist-target}}$ pointing from the wrist to the intended target: $\theta_{\text{hand}} = \min(\arccos\left(\frac{\vec{v}_{\text{wrist-base}} \cdot \vec{v}_{\text{wrist-target}}}{||\vec{v}_{\text{wrist-base}}|| ||\vec{v}_{\text{wrist-target}}||}\right), \arccos\left(\frac{\vec{v}_{\text{wrist-tip}} \cdot \vec{v}_{\text{wrist-target}}}{||\vec{v}_{\text{wrist-tip}}|| ||\vec{v}_{\text{wrist-target}}||}\right))$.

3. Arm Orientation Deviation (Figure 5 (c)) defined as the angular difference between the forearm vector $\vec{v}_{\text{elbow-wrist}}$ pointing from the elbow to the wrist and the the vector $\vec{v}_{\text{elbow-target}}$ pointing from the elbow to the target: $\theta_{\text{arm}} = \arccos\left(\frac{\vec{v}_{\text{elbow-wrist}} \cdot \vec{v}_{\text{elbow-target}}}{||\vec{v}_{\text{elbow-wrist}}|| \, ||\vec{v}_{\text{elbow-target}}||}\right)$.

For visual focus (Figure 5 (d)), the gaze angular deviation is computed by: $\theta = \arccos\left(\frac{\vec{v}_{\text{face}} \cdot \vec{v}_{\text{sight}}}{||\vec{v}_{\text{face}}|| \, ||\vec{v}_{\text{sight}}||}\right)$, where the facial orientation vector $\vec{v}_{\text{face}}$ points from the midpoint of the two ear joints to the midpoint of the eyes and nose. Meanwhile, line of sight vector $\vec{v}_{\text{sight}}$ extends from the eyes toward the intended target.

## C  ABLATION STUDY

### C.1  QUANTITATIVE EVALUATION ON BEATV2

To evaluate the generalizability of our framework beyond intent-driven scenarios, we compare the no-intent variant of our baseline against state-of-the-art (SOTA) models for conventional co-speech gesture generation. This experiment focuses on three core dimensions: gesture naturalness, diversity, and audio co-occurrence quality, without the guidance of explicit spatial intent targets.

**Experimental Setup.**  The no-intent baseline retains the speech-processing backbone and Transformer architecture but discards components specific to intent target (e.g., intent encoding and intent-aware attention module). We train and test our baseline on the BEAT Liu et al. (2022) dataset, which only involves the gesture of standing and talking straight ahead with no change in intention. The BEAT dataset contains 76 hours of multimodal speech data, including audio, transcripts, and full-body motion captured from 30 speakers performing in eight emotional styles and four different languages. We only use the speech data of English speakers 2. We use the FGD, Diversity and BC metrics for evaluaion. The calculation of FGD and Diversity and the bench value follow Liu et al. (2024), the calculation of BC and the bench value follow Qi et al. (2024).

**Quantitative Results**  As shown in the Figure 6, the no-intent variant of our baseline model still achieves good performance on the regular co-speech gesture generation task. It is better in matching the speech rhythm (highger BC) and is similar to EMAGE in terms of naturalness of the gesture movement (lower FGD).

Table 6: Quantitative evaluation on BEATv2 of the english speaker 2. We report FGD $\times 10^{-1}$, BC $\times 10^{-1}$, Diversity.

| Method | FGD $\downarrow$ | BC $\uparrow$ | Diversity $\uparrow$ |
|---|---|---|---|
| Trimodal | 12.41 | 0.75 | 7.724 |
| HA2G | 12.32 | 0.76 | 8.626 |
| CaMN | 6.644 | 0.82 | 10.86 |
| DiffStyleGesture | 8.811 | 0.81 | 11.49 |
| EMAGE | **5.512** | 0.85 | 13.06 |
| Ours | 6.020 | **0.87** | 10.38 |

### C.2  ABLATION STUDY ON MIXTURE TRAINING SCHEMES

To validate the effectiveness of our proposed mixture batch sampling strategy, we compare it against two common alternative training schemes. The detailed results are presented in Table 7. The schemes under comparison are:

1. **Batch Resampling (Ours)**: As described in the main paper, this strategy constructs each batch by drawing samples from Track-II and Track-I at a fixed 80/20 ratio, ensuring a consistent focus on the more challenging interactive data.

2. **Uniform Sampling**: This serves as a baseline where all data from the combined Track-I and Track-II datasets are placed in a single pool and sampled uniformly. This reflects a naive approach that does not account for the significant data imbalance between the two tracks.

3. **Two-Stage Training**: First, pre-train the model exclusively on the larger, more general Track-I dataset to learn a foundational understanding of spatially interactive gestures. Then, it is fine-tuned on the Track-II dataset, which contains explicit interactive semantic cues in audio, to specifically learn the critical task of synchronizing gesture interaction timing with the audio.

As demonstrated in Table 7, our batch resampling approach consistently yields the best performance across the majority of key metrics, particularly on the Track-II evaluation set, which directly measures the model's ability to generate precise, semantically timed interactive gestures. The uniform sampling approach struggles, especially on Track-II metrics like IOU and IAR, which confirms that the model fails to learn the crucial patterns from the underrepresented interactive examples.

Interestingly, the two-stage strategy also falls short of our proposed method. While this approach is often effective, we hypothesize that it suffers from a degree of catastrophic forgetting; during fine-tuning on the smaller Track-II dataset, the model may lose some of the valuable, generalized motion priors learned from the larger Track-I data. In contrast, our mixture sampling method provides a concurrent and balanced exposure to both data distributions within each training step. This forces the model to learn a unified representation that is both generalizable and specialized, proving to be the most effective strategy for this task.

Table 7: Results of ablation study on different mixture training schemes.

| | Intent Rep. | Visual Intent | | Pointing Intent | | |
|---|---|---|---|---|---|---|
| | | IAR@30°↑ | IOU@30°↑ | min IAD↓ | IAR@15°↑ | IOU@15°↑ |
| Track-I | batch resampling (ours) | 0.796 | - | 10.459 | 0.358 | - |
| | uniform sampling | 0.788 | - | 9.417 | 0.448 | - |
| | two-stage | 0.589 | - | 45.634 | 0.018 | - |
| Track-II | batch resampling (ours) | 0.710 | 0.545 | 5.153 | 0.846 | 0.637 |
| | uniform sampling | 0.685 | 0.469 | 7.317 | 0.661 | 0.509 |
| | two-stage | 0.733 | 0.544 | 5.203 | 0.814 | 0.611 |

## C.3 ABLATION STUDY ON MULTIMODAL FUSION SCHEMES

To assess the effectiveness of our multimodal fusion architecture, we investigate three distinct schemes for integrating the conditional inputs into the Diffusion Transformer backbone. Our model processes two categories of conditioning information: audio-related features (speech audio and transcripts) and intent-related features (target trajectories and intent category). The fusion schemes are as follows:

1. **Two-step Fusion (ours)**: This is our proposed method. Within each block of the DiT, we employ two separate cross-attention modules. The first module is dedicated to fusing the audio-related features with the gesture sequence, while the second independently fuses the intent-related features. This design allows the model to disentangle and specifically process the distinct spatial and temporal cues from each modality group.

2. **One-step Fusion**: In this variant, we combine all conditional features (both audio- and intent-related) into a single conditioning sequence. We then use only one cross-attention module per DiT block to fuse this unified context with the gesture sequence.

3. **Pre-fusion**: This baseline approach eschews dedicated cross-attention modules altogether. Instead, the embeddings of all conditional features are concatenated with the noisy gesture embeddings at the very input of the DiT. This simpler method integrates conditional information only once, before the main denoising process begins.

The results in Table 8 clearly demonstrate the superiority of our proposed two-step fusion mechanism. While the one-step fusion performs competitively, our two-step approach achieves better results on the most critical Track-II metrics (e.g., IAR, IOU), suggesting that allowing the model to handle audio and intent information through separate pathways leads to a more nuanced and accurate integration. The performance of the prefusion baseline is significantly lower, highlighting the importance of dynamically integrating conditional information throughout the diffusion process via cross-attention,

rather than relying on a static initial fusion. This confirms that our architectural choice to use distinct, specialized fusion modules is a key contributor to the model's overall performance.

Table 8: Results of ablation study on different multimodal fusion schemes.

| Intent Rep. | | Visual Intent | | Pointing Intent | | |
|---|---|---|---|---|---|---|
| | | IAR@30°↑ | IOU@30°↑ | min IAD↓ | IAR@15°↑ | IOU@15°↑ |
| Track-I | two-step fusion (ours) | 0.796 | - | 10.459 | 0.358 | - |
| | one-step fusion | 0.768 | - | 12.436 | 0.329 | - |
| | prefusion | 0.763 | - | 10.314 | 0.402 | - |
| Track-II | two-step fusion (ours) | 0.71 | 0.545 | 5.153 | 0.846 | 0.637 |
| | one-step fusion | 0.735 | 0.555 | 5.043 | 0.843 | 0.637 |
| | prefusion | 0.704 | 0.515 | 5.174 | 0.834 | 0.632 |

Table 9: Results of ablation study on different spatial representations of intent target. pink and lightpink indicate the best and the second best.

| Intent Rep. | | Visual Intent | | Pointing Intent | | |
|---|---|---|---|---|---|---|
| | | IAR@30°↑ | IOU@30°↑ | min IAD↓ | IAR@15°↑ | IOU@15°↑ |
| Track-I | global trans (ours) | 0.796 | - | 10.459 | 0.358 | - |
| | xz unit + y trans | 0.798 | - | 8.562 | 0.44 | - |
| | xz polar + y trans | 0.801 | - | 9.918 | 0.407 | - |
| | sphere coords | 0.791 | - | 11.304 | 0.373 | - |
| Track-II | global trans (ours) | 0.71 | 0.545 | 5.153 | 0.846 | 0.637 |
| | unit xz | 0.702 | 0.544 | 5.172 | 0.841 | 0.638 |
| | polar xz | 0.696 | 0.522 | 4.956 | 0.843 | 0.634 |
| | sphere | 0.731 | 0.553 | 5.109 | 0.838 | 0.633 |

## C.4 ABLATION STUDY ON SPATIAL REPRESENTATION OF INTENT TARGET

The representation of the intent target's spatial location is a critical input for guiding gesture generation. To identify a robust representation, we conducted an ablation study comparing several different coordinate systems. The goal was to find a balance between expressive power and learnability. The evaluated schemes include:

1. **Global Translation (Ours)**: Direct use of raw $(x, y, z)$ coordinates with the speaker's initial position as the origin, serving as the simplest spatial representation.

2. **XZ Unit Vector + Y-axis translation** : $(x_u, z_u, r, \log(r + 1), y)$. Decompose horizontal coordinates $(x, z)$ into a unit vector $(x_u, z_u)$ and distance $r = \sqrt{x^2 + z^2}$, combined with vertical height $y$ and $\log(r + 1)$ for nonlinear distance compression. This design emphasizes horizontal directional sensitivity while mitigating variance in long-distance perception, mimicking human spatial cognition that prioritizes horizontal orientation over absolute distance.

3. **XZ Polar Coordinates + Y-axis translation**: $(\theta, r, \log(r + 1), y)$. Encoding $(x, z)$ as polar $(\theta, r)$ with $\theta$ (azimuth angle) and $r$ (distance), plus $y$ and $\log(r + 1)$. This is another spatial representation that emphasizes the horizontal direction with distance robustness.

4. **Spherical Coordinates**: Converts Cartesian coordinates to spherical $(\theta, \phi, d, \log(d + 1))$, where $\theta$ (azimuth) and $\phi$ (elevation) capture directional angles, $d$ denotes Euclidean distance, and $\log(d + 1)$ for nonlinear compression of distance. Spherical coordinates is similar to the human perception of spatial directionality.

As the results in Table 9 indicate, no single representation demonstrated universal superiority across both the Track-I and Track-II datasets. While certain specialized representations, such as the spherical

coordinates, showed marginal benefits on specific metrics for the explicit interaction data in Track-II, their performance on the more general Track-I data was not as strong.

Ultimately, we selected the global translation (absolute XYZ coordinates) as our model's default representation. This choice was based on its consistent and strong performance across both datasets, as well as its simplicity and directness, which avoids introducing potentially complex inductive biases from other coordinate systems. This makes it a robust and reliable choice for the overall task.

