# OpenReview forum: "SIG-Chat: Spatial Intent-Guided Conversational Gesture Generation Involving How, When and Where"
_ICLR.cc/2026/Conference — ICLR 2026 Conference Withdrawn Submission_

### Official Review · Reviewer_mP3H · 2025-10-25

**Soundness:** 2
**Presentation:** 2
**Contribution:** 2
**Rating:** 2
**Confidence:** 5

**Summary:**

This paper presents a new dataset for spatial context–aware gesture generation. The dataset comprises over 7,000 clips (approximately 9 hours) collected using a motion capture system. The authors also propose a baseline method that incorporates an intention-aware encoder. In addition, they introduce new evaluation metrics, and the model with intention awareness demonstrates clear benefits, as further supported by user studies.

**Strengths:**

Incorporating spatial awareness into gesture generation is a promising direction. This paper presents a dataset and a corresponding benchmark to support research in this area. The dataset will be beneficial for the gesture generation community.

**Weaknesses:**

There are several major weaknesses:

1) The data collection process lacks transparency. It is not clear what kind of instructions were given to the participants. Moreover, the data collection is not grounded in the relevant literature, such as behavioural science, which raises concerns regarding potential bias.

2) Although the dataset contains over 7,000 clips, it was collected from only six participants, which significantly limits its generalisability. Gesture generation, in particular, is influenced by many personal factors such as culture and personality.

3) The scenarios appear more appropriate for two-person interactions, involving elements such as gaze and pointing. It is not clear why only a single person was recorded. (On a related note, Figure 2 depicts two people, which makes this aspect confusing.)

4) The motivation for including pointing gestures and gaze, as described in the paper, is not well justified, especially with regard to their transferability to humanoid robots. Similarly, squatting and lying down do not seem suitable for robotics applications. The paper lacks a convincing practical application. For example, it would have been more realistic if human–robot collaboration had been considered, where the robot could share gaze with a human or perform pointing gestures to achieve shared goals.

5) As acknowledged by the authors in Section 5.3, the dataset is biased.

Taken together, although this dataset may be beneficial, it focuses on a very niche problem with limited evidence of generalisability and insufficient grounding in the existing literature. Moreover, its practical applicability remains unclear.

**Questions:**

1) Please elaborate on the generalisability of this dataset. Why were only six participants included in the data collection process?
2) Was the data collection approved by an ethics committee? If so, please provide details of the approval process.
3) Please justify the choices made during data collection. How were the gaze and pointing gestures determined? What instructions were given to participants? Additionally, why was a single person recorded rather than an interactive or multi-person setting?

**Details Of Ethics Concerns:**

The paper does not specify whether the data collection process was approved by an ethics committee, which raises concerns. It is not even clear how these 6 participants were selected, whether diversity was taken into account in the recruitment.

---

### Official Review · Reviewer_jZrg · 2025-10-31

**Soundness:** 3
**Presentation:** 2
**Contribution:** 2
**Rating:** 4
**Confidence:** 3

**Summary:**

The paper describes a generation process of gestures used in dialogue.
The provided definitions make the description self-consistent.
The related work section contains a lot of references with  brief descriptions.
Additionally, the principal quantities for the metrics are introduced. The global architecture is described in figure 3. The blocks of the elaboration are shortly described. Probably a longer description would make the architecture easier to reproduce.
The experiments show quantitative results with defined metrics.

In Table 2, the metrics for human interactions are reported. They are also called ground-truth. This label is misleading, and a label just referring to human performance should be used.
Also, in Table 3, the results are compared with the Ground Truth.  It is strange that the proposed method shows performances that are better than the GT.


In section 6.4, the last sentence is “Thank you again for helping us improve the paper’s adequacy of assessment. “
The paper presents an interesting dataset in the field of human-robot interaction. Some portions of the denoising network are shown but are not deeply described.
As a minor issue, for all the used software and hardware, a reference should be indicated for their description (e.g., Xsens, HTC Vive, MVN software). The more information are provided the more replicable is the experimental setup.

**Strengths:**

The paper describes a dataset of gestures used in dialogue.

Generally, content is clear, and the provided definitions make the description self-consistent.

**Weaknesses:**

Some portion of the architecture, used as denoising network are not described in detail. Also, if the network components are trained from scratch or a fine tuning has been used was not declared.
The comparisons with other models are missing

**Questions:**

How the network in figure 3 has been trained?

Can your approach be compared, in a qualitive and quantitive way with other similar techniques?

---

### Official Review · Reviewer_c9jB · 2025-10-31

**Soundness:** 2
**Presentation:** 2
**Contribution:** 2
**Rating:** 4
**Confidence:** 2

**Summary:**

The paper introduces SIG-Chat, a multimodal speech–gesture dataset with explicit intent annotations and 3D spatial metadata of intent targets to support intent-aware conversational gesture synthesis; presents a baseline model that fuses audio, transcript, initial posture description, and the target’s trajectory to generate spatially interactive co-speech gestures; and establishes a benchmark with three metrics that explicitly evaluate temporal alignment and spatial interaction accuracy driven by multimodal inputs.

**Strengths:**

The paper presents a dataset for modeling conversational human behavior that, unlike prior datasets, integrates additional modalities—including spatial intent—to better capture interactive dynamics.

**Weaknesses:**

- The dataset adds additional modalities, but it is unclear whether its scale are sufficient for a benchmark. Although it expands to four modalities (where prior work often uses one or two), the overall data volume appears smaller than some single-modality datasets, raising overfitting concerns. The benchmark design may also introduce bias and hinder fair evaluation.

- Data quality is not well substantiated: there are no qualitative or quantitative comparisons to existing datasets or models. Beyond annotation/caption quality, it is unclear whether the scenarios genuinely reflect typical, real-world conversational behavior.

- Facial features are a core modality for understanding conversational behavior; while prior work includes them, it is unclear whether the proposed dataset and model incorporate this modality.

**Questions:**

Is facial feature also included in this dataset? If it is, please explain the process of data collection and quality?

---

### Official Review · Reviewer_bcnn · 2025-11-01

**Soundness:** 2
**Presentation:** 2
**Contribution:** 2
**Rating:** 4
**Confidence:** 4

**Summary:**

This paper addresses a main limitation in co-speech gesture generation: the inability of current models to control when and where a gesture interacts with the environment. To solve this, the authors introduce a new large-scale motion capture dataset (SIG-Chat) with synchronized 3D spatial intent data, novel metrics to evaluate the spatial and temporal accuracy of these gestures, and a Diffusion Transformer baseline model that generates gestures guided by audio, text, and 3D target locations. The work is validated with quantitative results, a user study, and a proof-of-concept deployment on a humanoid robot.

**Strengths:**

- The paper tackles the important and previously overlooked problem of generating spatially-aware conversational gestures.
- The model significantly outperforms its ablated baseline ("w/o intent") in both quantitative evaluations and a user study.

**Weaknesses:**

- The model is given the entire 3D target trajectory as input. This means it doesn't need to infer where or when to point; this information is provided. This limits its practical use in dynamic, unscripted scenarios.
- The model processes the entire audio and trajectory sequence at once to generate the full gesture clip. This offline approach is not suitable for real-time, interactive applications, and the robot demo appears to be a playback of a pre-computed motion.
- The primary comparison is an ablation of their own model. The paper does not adapt other state-of-the-art gesture generation models (e.g., winner models from the GENEA challenge) to this new task, which would have provided a stronger context.

Minors:
- Typo: Figure 3, Sacle -> scale
- Line 476. Text from previous rebuttal.

**Questions:**

- How is the dataset compared and related to the GENEA challenge?
- Could you elaborate on the data collection process? Was the dialogue scripted for the actors? Were they given specific instructions like "now say your line and point to the target," or was it more improvisational? This would help in understanding the naturalness and spontaneity of the collected gestures and speech.

---

### Note · Authors · 2025-11-18

I have read and agree with the venue's withdrawal policy on behalf of myself and my co-authors.